# Synthesis and Evaluation of 11-Butyl Matrine Derivatives as Potential Anti-Virus Agents

**DOI:** 10.3390/molecules27217563

**Published:** 2022-11-04

**Authors:** Wanjun Ni, Lizhong Wang, Hongjian Song, Yuxiu Liu, Qingmin Wang

**Affiliations:** State Key Laboratory of Elemento-Organic Chemistry, Research Institute of Elemento-Organic Chemistry, College of Chemistry, Frontiers Science Center for New Organic Matter, Nankai University, Tianjin 300071, China

**Keywords:** matrine, anti-TMV activity, fungicidal activity, insecticidal activity, structure-activity relationship

## Abstract

Matrine derivatives were reported to have various biological activities, especially the ester, amide or sulfonamide derivatives of matrine deriving from the hydroxyl or carboxyl group at the end of the branch chain after the D ring of matrine is opened. In this work, to investigate whether moving away all functional groups from the C-11 branch chain could have an impact on the bioactivities, such as anti-tobacco mosaic virus (TMV), insecticidal and fungicidal activities, a variety of *N*-substituted-11-butyl matrine derivatives were synthesized. The obtained bioassay result showed that most *N*-substituted-11-butyl matrine derivatives had obviously enhanced anti-TMV activity compared with matrine, especially many compounds had good inhibitory activity close to that of commercialized virucide Ningnanmycin (inhibition rate 55.4, 57.8 ± 1.4, 55.3 ± 0.5 and 60.3 ± 1.2% at 500 μg/mL; 26.1, 29.7 ± 0.2, 24.2 ± 1.0 and 27.0 ± 0.3% at 100 μg/mL, for the in vitro activity, in vivo inactivation, curative and protection activities, respectively). Notably, *N*-benzoyl (**7**), *N*-benzyl (**16**), and *N*-cyclohexylmethyl-11-butyl (**19**) matrine derivatives had higher anti-TMV activity than Ningnanmycin at both 500 and 100 μg/mL for the four test modes, showing high potential as anti-TMV agent. Furthermore, some compounds also showed good fungicidal activity or insecticidal activity.

## 1. Introduction

Matrine (Figure 1) and its homologous alkaloids are mainly isolated from the dried root of *Sophora flavescens*, *Sophora alopecuroides*, and *Sophora tonkinensis* [1,2]. As matrine has a broad scope of biological activities, lots of research has reported on its structural optimization and activity evaluation. Matrine is a fused tetracyclic compound composed of two quinolinone skeletons. There are two types of artificial matrine derivatives which are basically obtained by semi-synthesis of matrine. One type is to retain the tetracyclic structure of matrine, and at the same time modify it at the C-13, 14 or 15 positions or fuse an additional ring on the D ring; the other is to open the D ring via amide hydrolysis and then introduce various substituents both on the nitrogen and the side chain at the C-11 position. Both types of derivatives showed good bioactivity. Matrine-family alkaloids and their modified derivatives have been widely studied in medical field, and the pharmacological activities and mechanism of action have been summarized in several reviews [3,4,5,6,7]. Moreover, many matrine derivatives are also found to have good agricultural activities [8,9], which is our concern. Xu group synthesized a large amount of matrine derivatives and found some of them had good insecticidal/acaricidal activity against *Mythimna separata* Walker, *Plutella xylostella* Linnaeus, *Aphis citricola* Van der Goot and/or *Tetranychus cinnabarinus*; these molecules included ether derivatives [10], oximinoether derivatives [11], imine derivatives [12], spiro-1,2,4-oxadiazoline-fused matrine derivatives [13], and pyrazolomatrine derivatives [14], etc., that retain the tetracyclic structure, as well as the acid/alcohol/ester derivatives [15,16] and amide derivatives [17,18] with an opened D-ring. Jiang and Qin jointly reported the synthesis and insecticidal activity evaluation of two series of tetracyclic matrine derivatives with halopyrazole motif [19] or cyclohexylamine group [20] both at the C-13 position. Huang et al. [21] prepared a series of C-14 arylmatrine derivatives and found they showed enhanced insecticidal effects against *Spodoptera exigua* Hubner.

Research and development of highly effective antiviral agents with natural products as lead compounds is a project being carried out by our research team [22,23,24,25,26]. In our previous work, we first reported matrine has higher antiviral activity against tobacco mosaic virus (TMV) than commercialized ribavirin, while the esters of 14-hydroxymethyl-15-deoxymatrines exhibit even better anti-TMV activity, and some of them also have a broad-spectrum fungicidal activity and insecticidal activity [27]. In our recent work, a series of matrine derivatives were prepared by opening the D ring and introducing acyl hydrazone moiety to the C-11 position of matrine, and those compounds were found to exhibit very good anti-TMV activity [28]. We proposed that the hydrogen bond donor and acceptor property of acyl hydrazone played an important role. In order to investigate whether the anti-TMV would be affected when changing the functional group to straight-chain alkyl group at the C-11 position, in this work, a variety of *N*-substituted-11-butyl matrine derivatives (Figure 1) were synthesized. The prepared compounds were evaluated for their anti-TMV activity, and the structure-activity relationship was systematically studied. In addition, their insecticidal activity and fungicidal activity were also investigated.

## 2. Results and Discussion

### 2.1. Synthesis

To prepare the C-11 butyl compounds, matrine was firstly treated with excessive potassium hydroxide to obtain the D-ring opened compound **A** by amide hydrolysis. Then, compound **A**, triethylamine and slightly excessive Boc anhydride were refluxed in methanol for 4 h to afford Boc-protected matrinic acid **B [29]**. The acid **B** was reduced to corresponding alcohol **C** by LiAlH_4_ in THF, which continued to react with sulfonyl chloride to give methanesulfonate **D**. Treated with LiAlH_4_, the matrinic methanesulfonate **D** was reduced to *N*-Boc-11-butyl matrine derivative **E**. The protecting group could be removed in excess concentrated hydrochloric acid at room temperature for 3 h to give compound **F**. With 11-butyl matrine derivative **F** well prepared, it was reacted with corresponding sulfonyl chlorides or acyl chlorides to form *N*-sulfonyl-11-butyl matrine derivatives (**1**–**5**) and *N*-acyl-11-butyl matrine derivatives (**6**–**15**), respectively (Figure 1). Moreover, reducing corresponding *N*-acyl-11-butyl matrine derivatives could provide *N*-alkyl-11-butyl matrine derivatives (**16**–**19**) (Figure 2). In addition, **F** reacted with isocyanate afforded urea-type matrine derivatives **20** and **21** (Figure 3).

Single-crystal X-ray diffraction of the hydrochloride salt of compound **15** (data were deposited at the Cambridge Crystallographic Data Centre (CCDC) with deposition number 1531045) confirmed the correctness of structure (Figure 2). It can be seen from the figure that the D ring of matrine has been opened, and the remaining three rings all adopt the chair configuration, and the chiral centers (C-5, C-6, C-7 and C-11) all maintain the original chirality structure. When the salt is formed, the newly-added hydrogen on the nitrogen is in the opposite direction to that of the C-6 hydrogen, suggesting it is the dominant conformation.

### 2.2. Biological Assay

#### 2.2.1. Antiviral Activity

The synthesized 11-butyl matrine derivatives were assayed for the antiviral activity against TMV in four processing modes: in vitro mode, in vivo inactivation mode, curative mode, and protection mode, with commercial Ribavirin and Ningnanmycin as standards (Table 1).

For the anti-TMV activities at the concentration of 500 µg/mL in vitro, almost all 11-butyl matrine derivatives were higher than ribavirin (inhibition rate 40.9%) and matrine (33.7%) except *N*-cyclopropanesulfonyl matrine derivative **1** and *N*-(2-fluorobenzenesulfonyl) derivative **6**. The activities of *N*-benzoyl (**7**) (62.4%), *N*-n-hexyl (**18**) (66.9%), *N*-cyclohexylmethyl (**19**) (63.1%) 11-butyl matrine derivatives, and *N*′-cyclohexyl urea-type 11-butyl matrine derivative (**20**) (68.4%) were much higher than ningnanmycin (55.4%), while the *N*-vinylsulfonyl (**4**), *N*-benzyl (**16**), *N*-nathphthyl matrine derivative (**17**) and *N*′-(3-methyl)phenyl urea-type matrine derivative (**21**) had similar activity to ningnanmycin.

Inspired by the in vitro activity, we then test for their in vivo anti-TMV activities. As expected, several compounds had higher activities than or equal to ningnanmycin (inhibition rates of 57.8 ± 1.4, 55.3 ± 0.5 and 60.3 ± 1.2% for inactivation, curative and protection effect, respectively, at 500 µg/mL), especially the compounds having good anti-TMV activities in vitro. Among *N*-sulfonyl compounds **1**–**5**, the in vivo activity of *N*-vinylsulfonyl compound **4** (62.3 ± 0.8, 58.7 ± 1.5, 65.9 ± 3.0%, 500 µg/mL) was best. Among *N*-(substituted)benzoyl-11-butyl matrine derivatives **6**–**8**, *N*-benzoyl derivative (**7**) (65.9 ± 2.0, 61.4 ± 3.0, 67.4 ± 2.3%, 500 µg/mL) showed better anti-TMV activity than the 2-fluorobenzoyl derivative (**6**) and the 4-methylbenzoyl derivative (**8**). Among the *N*-fatty acyl matrine derivatives **10**–**15**, *N*-*n*-hexylformyl matrine derivative with a long alkyl chain (**12**) (63.8 ± 2.4, 57.9 ± 3.5, 61.0 ± 4.1%, 500 µg/mL) were much better than the others. All four *N*-alkyl-11-butyl matrine derivatives **16**–**19** exhibited excellent anti-TMV activity in vivo. Compared with their corresponding *N*-acyl substrates (**7**, **9**, **12** and **15**), **16** and **18** had similar activity to **7** and **12**, while **17** and **19** had better activity than **9** and **15**. Among them, *N*-cyclohexylmethyl-11-butyl matrine derivative (**19**) (70.1 ± 1.0, 66.4 ± 3.5, 72.3 ± 0.9%, 500 µg/mL) exhibited the best, much higher than Ningnanmycin. Two urea-type matrine derivatives **20** and **21** both had good anti-TMV activities, with the alkyl urea **20** (61.5 ± 3.5, 62.1 ± 2.4, 67.0 ± 1.5%, 500 µg/mL) better than aryl urea **21** (52.8 ± 0.5, 57.6 ± 2.3, 59.0 ± 1.0%, 500 µg/mL).

Briefly, the order of the anti-TMV activities is *N*-alkyl-11-butyl > *N*-sulfonyl-11-butyl > *N*-acyl-11-butyl matrine derivatives. The *N*-cyclohexylmethyl-11-butyl matrine derivative **19** is the most potent anti-TMV agent among the synthesized compounds, and it is comparable with our previously prepared matrine derivatives bearing an acylhydrazone moiety at the C-11 butane chain (4-bromine-3-indoleformaldehyde *N*-benzylmatrine-11-butanehydrazone; 71.8 ± 2.8, 66.8 ± 1.3, 69.5 ± 3.1%, 500 µg/mL, [28]). This indicates both the substituents at the nitrogen and functional groups at the 11-alkyl chain play important roles for the activity. To find more potent compounds, *N*-alkyl derivatives require further research, including expanding the compound library and investigating the anti-TMV mechanism.

#### 2.2.2. Insecticidal Activities

Matrine and its derivatives all evaluated for the insecticidal/acaricidal activities against seven kinds of insects/mites, including oriental armyworm (*Mythimna separate*), cotton bollworm (*Helicoverpa armigera*), corn borer (*Ostrinia nubilalis*), diamond back moth (*Plutella xylostella*), aphid (*Aphis medicnginis* Koch), mosquito larvae (*Culex pipiens pallens*) and spider mite (*Tetranychus cinnabarinus*). Bioassay showed that these 11-butyl derivatives showed good insecticidal selectivity on diamond back moth. Half of the compounds exhibited more than 60% mortality against diamond back moth at 600 µg/mL, equivalent to the level of *N*-benzylmatrine-11-butanehydrazones. To other species, most of the compounds exhibited lower insecticidal activity, only *N*-cyclohexylmethyl-11-butyl matrine (**19**) and *N*-[12-(11-butyl matrine)]-*N*′-cyclohexyl urea (**20**) exhibited more than 70% mortality against oriental armyworm and cotton bollworm at 600 µg/mL, as well compound **20** and *N*-[12-(11-butyl matrine)]-*N*′-3-methylphenyl urea (**21**) exhibited 100% mortality against corn borer. All compounds had little activity against aphids and adult mites, which is consistent with most of *N*-benzylmatrine-11-butanehydrazones. (The total bioassay data of **1**–**21** can be found in the Appendix A.)

#### 2.2.3. Fungicidal Activity

Matrine and their derivatives all evaluated for fungicidal activities against 14 kinds of plant pathogens (*Fusarium oxysporum sp*. *cucumeris*; *Cercospora arachidicola Hori*; *Physalospora piricola*; *Rhizoctonia cerealis*; *Bipolaris maydis*; *Colletotrichum orbiculare*; *Fusarium moniliforme*; *Alternaria solani*; *Fusarium graminearum*; *Phytophthora infestans*; *Phytophthora capsici*; *Sclerotinia sclerotiorum*; *Botrytis cinerea*; *Rhizoctonia solani*) by mycelial growth method. Partial data are listed in Table 2, and the total fungicidal data can be found in the Appendix A.

Generally, these compounds do not have outstanding fungicidal activity, and only individual compounds show certain fungicidal ability against individual strains. Specifically, at 50 mg/kg, *N*-pyridyl-3-sulfonyl-11-butyl matrine (**3**) exhibited more than 60% inhibitory activity against *Physalospora piricola* and *Rhizoctonia cerealis*; *N*-*n*-hexylmethyl-11-deoxymatrine (**18**) exhibited more than 60% inhibitory activity against *Cercospora arachidicola Hori*, *Physalospora piricola*, *Rhizoctonia cerealis*, and *Fusarium moniliforme* at the same concentration; and *N*-[12-(11-butyl matrine)]-*N*′-3-methylphenyl urea (**21**) exhibited more than 70% against both *Phytophthora capsici* and *Sclerotinia sclerotiorum*. No apparent selectivity and structure-activity relationship was found.

## 3. Materials and Methods

### 3.1. Instruments and Chemicals

Matrine was purchased from Baoji Biological Development Co. Ltd. Reagents were purchased from commercial sources and were used as received. All anhydrous solvents were dried and purified by standard techniques just before use. Reaction progress was monitored by thin-layer chromatography on silica gel GF_254_ with detection by UV. Melting points were determined using an X-4 binocular microscope melting point apparatus, and the thermometer was uncorrected. ^1^H and ^13^C NMR spectra were obtained by using Bruker AV 400 with CDCl_3_ or DMSO-*d*_6_ containing 0.03% tetramethylsilane (TMS) as solvent. Chemical shifts (*δ*) were given in parts per million (ppm) and were measured downfield from internal TMS. High-resolution mass spectra (HRMS) were obtained with an FT-ICR MS spectrometer (Ionspec, 7.0 T).

### 3.2. General Synthesis

The synthetic routes are depicted in Figure 1, Figure 2 and Figure 3. Compound **A** and **B** were synthesized referring to literature [29].

#### 3.2.1. Synthesis of *N*-t-Butyloxycarbonyl-11-(4-hydroxybutyl) Matrine Derivative (**C**)

To a suspension of LiAlH_4_ (0.83 g, 21.80 mmol) in THF (40 mL), **B** (4.00 g, 10.90 mmol) was added in one portion in an ice bath. The reaction mixture was stirred at 0 °C for 10 min, refluxed for another 10 h, and then quenched with water. After the evaporation of THF, the aqueous layer was extracted with ethyl acetate three times. The combined organic phase was washed with brine, dried over MgSO_4_, filtered, and concentrated under reduced pressure. The resulting residue was purified by column chromatography (V_DCM_:V_MeOH_ = 10:1) to give compound **C** (2.90 g, yield 63%) as yellow oil.

#### 3.2.2. Synthesis of Methanesulfonate of *N*-t-Butyloxycarbonyl-11-(4-hydroxybutyl) Matrine Derivative (**D**)

Triethylamine (3.40 g, 24.60 mmol) was added to the mixture of compound **C** (4.00 g, 11.30 mmol) in freshly distilled dichloromethane. Methylsulfonyl chloride (1.49 mL, 19.20 mmol) was slowly added dropwise in an ice bath, and then the mixture was stirred overnight at room temperature. The mixture was quenched by saturated sodium carbonate solution and extracted with dichloromethane. The combined organic phase was concentrated under reduced pressure, and the resulting residue was purified by column chromatography (EA) to give compound **D** (3.80 g, yield 84%) as yellow oil.

#### 3.2.3. Synthesis of *N*-t-Butyloxycarbonyl-11-butyl Matrine Derivative (**E**)

To a suspension of LiAlH_4_ (0.84 g, 22.20 mmol) in THF (40 mL), **D** (4.50 g, 11.10 mmol) was added in one portion in an ice bath. The reaction mixture was stirred at 0 °C for 10 min, refluxed for another 10 h, and then quenched with water (30 mL). After evaporation of THF, the aqueous layer was extracted with ethyl acetate (20 mL × 3). The combined organic phase was washed with brine, dried over MgSO_4_, filtered, and concentrated under reduced pressure. The resulting residue was purified by column chromatography (V_DCM_:V_MeOH_ = 20:1) to give compound **E** (3.00 g, yield 81%) as yellow oil.

#### 3.2.4. Synthesis of *N*-H-11-Butyl Matrine Derivative (**F**)

To compound **E** (2.50 g, 6.60 mmol) in 1,4-dioxane (10 mL), concentrated hydrochloric acid (10.00 mL, 78.20 mmol) was added dropwise. The reaction mixture was stirred at room temperature for 6 h. After evaporation of the solvent, saturated potassium carbonate solution was added to make pH = 10, then the aqueous layer was extracted with ethyl acetate (20 mL × 3). The combined organic phase was washed with brine, dried over MgSO_4_, filtered, and concentrated under reduced pressure to give compound **F** (1.20 g, yield 80%) as yellow oil, which solidified in air. Mp 46–47 °C. ^1^H NMR (400 MHz, CDCl_3_) *δ* 3.25 (t, *J* = 12.0 Hz, 1H), 3.09–3.03 (m, 1H), 2.80 (dd, *J* = 16.0, 12.0 Hz, 2H), 2.63 (dd, *J* = 12.0, 4.0 Hz, 1H), 2.08 (s, 1H), 1.97–1.88 (m, 3H), 1.75–1.51 (m, 6H), 1.44–1.16 (m, 9H), 0.90 (t, *J* = 8.0 Hz, 3H). ^13^C NMR (100 MHz, CDCl_3_) *δ* 63.7, 56.7, 56.6, 51.0, 45.5, 40.8, 36.3, 31.8, 27.3, 26.5, 25.8, 22.0, 20.7, 20.3, 13.0. HRMS (ESI) calculated for [C_15_H_28_N_2_+H]^+^ 237.2325, found 237.2330.

#### 3.2.5. Synthesis of *N*-Cyclopropyl Sulfonyl-11-butyl Matrine Derivative (**1**)

Triethylamine (0.67 mL, 4.80 mmol) was added to compound **F** (0.38 g, 1.60 mmol) in dried CH_2_Cl_2_ (10 mL). After the mixture was stirred for 5 min, cyclopropanesulfonyl chloride (0.33 mL, 3.20 mmol) was added, and then the mixture was stirred at room temperature for 12 h. After adding a solution of 20% sodium hydroxide (20 mL), the organic phase was separated and the aqueous layer was extracted with dichloromethane (20 mL × 3). The combined organic phase was washed with brine, dried over MgSO_4_, filtered, and concentrated under reduced pressure. The resulting residue was purified by column chromatography (EA as eluent) to give compound **1** (0.30 g, yield 53%) as yellow oil. ^1^H NMR (400 MHz, CDCl_3_) *δ* 3.72 (dd, *J* = 12.0, 8.0 Hz, 1H), 3.42 (d, *J* = 8.0 Hz, 2H), 2.77 (t, *J* = 12.0 Hz, 2H), 2.50–2.44 (m, 1H), 2.10–1.17 (m, 22H), 1.02–0.96 (m, 1H), 0.92 (t, *J* = 8.0 Hz, 3H). ^13^C NMR (100 MHz, CDCl_3_) *δ* 63.5, 57.9, 56.9, 47.7, 39.7, 34.7, 31.3, 30.1, 28.3, 28.0, 27.9, 22.9, 21.0, 20.9, 14.1, 5.9, 5.1. HRMS (ESI) calculated for [C_18_H_32_N_2_O_2_S+H]^+^ 341.2257, found 341.2260.

Compounds **2**–**15** were prepared by reacting **F** with corresponding sulfonyl chloride or acyl chloride using a synthetic procedure similar to that of compound **1**. The yields, melting points, ^1^H and ^13^C NMR data, and HRMS data are provided as Appendix A.

#### 3.2.6. Synthesis of *N*-Benzyl-11-butyl Matrine Derivative (**16**)

*N*-benzoyl-11-butyl matrine derivative (0.60 g, 1.70 mmol) was added to a suspension of LiAlH_4_ (0.13 g, 3.4 mmol) in THF (40 mL) in small portion in an ice bath. The reaction mixture was stirred at 0 °C for 10 min, refluxed for another 10 h, and then quenched with water (30 mL). After evaporation of THF, the aqueous layer was extracted with ethyl acetate (20 mL × 3). The combined organic phase was washed with saturated brine, dried over MgSO_4_, filtered, and concentrated under reduced pressure. The resulting residue was purified by column chromatography (V_DCM_:V_MeOH_ = 20:1) to give compound **16** (0.30 g, yield 79%) as white oil. ^1^H NMR (400 MHz, CDCl_3_) *δ* 7.34–7.20 (m, 5H), 4.09 (d, *J* = 12.0 Hz, 1H), 3.04 (d, *J* = 12.0 Hz, 1H), 2.83–2.75 (m, 2H), 2.55 (t, *J* = 12.0 Hz, 1H), 2.32–2.28 (m, 1H), 2.01 (s, 1H), 1.96–1.23 (m, 19H), 0.87 (t, *J* = 4.0 Hz, 3H). ^13^C NMR (100 MHz, CDCl_3_) *δ* 128.8, 128.5, 128.1, 126.9, 126.5, 64.6, 57.6, 57.4, 57.0, 52.2, 38.5, 34.2, 29.7, 29.2, 28.2, 27.4, 26.2, 23.2, 21.6, 21.4, 14.1. HRMS (ESI) calculated for [C_22_H_34_N_2_+H]^+^ 327.2795, found 327.2798.

Compounds **17**–**19** were prepared similarly by using a synthetic procedure similar to that of compound **16**. The yields, melting points, ^1^H and ^13^C NMR data, and HRMS data are provided as Appendix A.

#### 3.2.7. Synthesis of *N*-[12-(11-Butyl Matrine)]-*N*′-cyclohexyl Urea (**20**)

Triethylamine (0.15 mL, 1.11 mmol) was added to compound **F** (0.24 g, 1.00 mmol) in dried CH_2_Cl_2_ (10 mL). After the mixture was stirred for 5 min, cyclohexyl isocyanate (0.13 mL, 1.05 mmol) was injected by syringe under argon, and then the mixture was stirred at room temperature for 12 h. After adding a solution of 20% sodium hydroxide (20 mL), the organic phase was separated and the aqueous layer was extracted with dichloromethane (20 mL × 3). The combined organic phase was washed with brine, dried over MgSO_4_, filtered, and concentrated under reduced pressure. The resulting residue was purified by column chromatography (V_DCM_:V_MeOH_ = 20:1) to give compound **20** (0.24 g, 67%) as a white solid. Mp 86–87 °C. ^1^H NMR (400 MHz, CDCl_3_) *δ* 4.35 (d, *J* = 8.0 Hz, 1H), 3.72–3.60 (m, 2H), 3.52 (dd, *J* = 12.0, 4.0 Hz, 1H), 3.19 (dd, J = 8.0, 4.0 Hz, 1H), 2.74 (d, *J* = 4.0 Hz, 2H), 2.04 (s, 1H), 1.95–1.07 (m, 28H), 0.89 (t, *J* = 4.0 Hz, 3H). ^13^C NMR (100 MHz, CDCl_3_) *δ* 158.3, 63.6, 56.8, 55.3, 48.9, 46.8, 39.1, 34.8, 34.1, 33.9, 32.2, 28.7, 28.5, 28.1, 25.7, 25.0, 25.0, 22.8, 21.2, 21.2, 14.1. HRMS (ESI) calculated for [C_22_H_39_N_3_O+H]^+^ 362.3166, found 362.3164.

Compound **21** was prepared similarly by reacting **F** with corresponding isocyanate using a synthetic procedure similar to that of compound **20**. The yields, melting points, ^1^H and ^13^C NMR data, and HRMS data are supplied as Appendix A.

### 3.3. Biological Assay

Detailed bioassay procedures for the anti-TMV, fungicidal, and insecticidal activities are described in our published literature [27] and can also be found in the Appendix A. According to statistical requirements, each bioassay was repeated at least three times except anti-TMV activity assay in vitro.

## 4. Conclusions

In brief, after opening up the D ring of matrine, reducing the C-11 side chain to butyl, and introducing sulfonyl, acyl, alkyl or carbamoyl to the nitrogen, we obtained several matrine derivatives and evaluated their anti-TMV, insecticidal and fungicidal activities. Bioassay showed that these compounds exhibited very good anti-TMV activities both in vitro and in vivo, especially the *N*-alkyl-11-butyl derivatives which were better than the compounds with other substituents on the nitrogen, and *N*-cyclohexylmethyl-11-butyl matrine derivative **19** exhibited the best activity both in vitro and in vivo, deserving extensive research. Compared with our previously prepared hydrazone derivatives of matrine, both the substituents on the nitrogen and the side chain affect their anti-TMV activity. Structure modification and anti-TMV mechanism studies will be carried out and reported in the future.

## Data Availability

Not applicable.

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
