# Peer review of "Synthesis and Evaluation of 11-Butyl Matrine Derivatives as Potential Anti-Virus Agents"

_molecules, 2022, doi:10.3390/molecules27217563_

Round 1

Reviewer 1 Report

In this manuscript, the authors presented the Synthesis and evaluation of 11-butyl matrine derivatives as potential antiviral agents which are particularly proven to be potent against tobacco mosaic virus (TMV). The authors also tested the synthesized molecules for insecticidal and fungicidal activities which were also found to show good fungicidal activity or insecticidal activity.

The authors disclosed a lot of interesting insights of matrin and their derivatives. The manuscript was nicely written and presented with all the necessary information. All compounds are well characterized.

Revise the table of contents part in the supporting information. The bioassay experimental information of TMV is missing in it and page numbers should be included appropriately.

It may also make more sense if the lead molecules of previously reported (reference 7) compared with current lead molecules for anti TMV activity.

Over all, this manuscript can be publishable in Molecules.

Author Response

Thanks for giving valuable advices to improve the manuscript quality. According to the suggestion, the introduction has been rewritten, now it mainly focuses on the structure types of matrine derivatives and corresponding agricultural activity. The discussion on the structure-activity relationship anti-TMV was revised, and the activity comparison between the current lead molecular and our previously reported molecular was also added. Many relevant research and references were added including two latest publications of this journal. The style of manuscript and references was also checked and uniformed. I believe that the revised version meets the standard of journal. If there is still need for improvement, please point out.

Reviewer 2 Report

This article described the design and synthesis of 11-butyl matrine derivatives, and their potential anti-virus, insecticidal and fungicidal activities have been evaluated, which indicated these matrine derivatives could be used as potential anti-TMV agents. Generally, this work is carefully designed and carried out, however, some minor issues should be concerned.

-Standard error for the in vitro inhibition rate in Table 1 is missing.

-The discussion on the biological activity should be improved, which is only a description of data, and a detailed analysis of the relationship between structure and activity should be added.

-The author doesn't identify the protons and carbons in 1H and 13C-NMR data. This will not be helpful for the reader to distinguish between the protons and carbons for the new compounds.

-Please label all the references in correct style, and check them carefully.

Author Response

Thanks for giving valuable advices to improve the manuscript quality. According to the suggestion, the discussion on the structure-activity relationship anti-TMV was revised, and the activity comparison between the current lead molecular and our previously reported molecular was also added. According to another reviewer’s suggestion, the introduction section has been rewritten. Several references were added, and the style of manuscript and references was checked and uniformed.

Note to mention that, the in vitro inhibition of anti-TMV we tested only once to see if these series of molecules deserving further in vivo test, so there is no standard error for vitro inhibition. This is indicated in the footnote of Table 1. Precise identification of protons and carbon in 1H and 13C-NMR data requires several 2D spectra. Therefore, I dare not give accurate attribution easily to avoid misleading others.

Reviewer 3 Report

Manuscript presented by Wanjun Ni et al. shows a study about synthesis and evaluation of 11-butyl matrine derivatives. An already well written and prepared manuscript. Easy to read and follow. I recommend the article to publish in Molecules journal but first the paper should be improve. My decision – Reconsider after minor revision. Comments to be considered, in order to further improve the manuscript quality:

(1)   Introduction need to be re-written, the information flow of introduction is poor. Focus on application and other method of synthesis / isolating of presented compounds.

(2)   Some sentences, like “Structurally, matrine alkaloids are fused tetracyclic compounds composed of two quinolinone skeletons”, need references.

(3)   Avoid lumping references as in [8-16] and all other. Instead summarise the main contribution of each referenced paper in a separate sentence.

(4)   “Material and methods” section should be improved. More information about analysis equipment (eg. NMR internal standard ect.).

(5)   Avoid abbreviations like “compd

(6)   Superscripts and subscripts as well as commas and periods should be change and correct. Avoid extra spaces and enters.

(7)   In order to show that the topic of presented manuscript is proper for the publication in MOLECULES please include into reference other publications of this journal. Manuscript also needs more latest references.

(8)   The style of manuscript (especially reference) should be improve (see template) eg. line181 and other.

(9)   The English correction is necessary.

Author Response

Thanks for giving valuable advices to improve the manuscript quality. According to the suggestion, the activity comparison between the current lead molecular and our previously reported molecular was added. The table of contents part in the supporting information was revised, and the bioassay experimental information of TMV was added. Furthermore, according to other reviewers’ suggestion, the reduction was rewritten and the discussion on the structure-activity relationship anti-TMV was also revised.